# Strain Concentration Ratio Analysis of Different Waterproofing Materials during Concrete Crack Movement

**DOI:** 10.3390/ma14164429

**Published:** 2021-08-07

**Authors:** Kyu-hwan Oh, Soo-yeon Kim

**Affiliations:** Institute of Construction Technology, Seoul National University of Science & Technology, 232 Gongneung-ro, Nowon-gu, Seoul 01811, Korea; kyuhwan.oh@seoultech.ac.kr

**Keywords:** waterproofing material, strain analysis, strain concentration ratio, new evaluation method

## Abstract

When a crack occurs under an installed waterproofing material and moves due to environmental effects (freeze–thaw, settlement, vibration, dead load, etc.), waterproofing materials without adequate elongation or tensile strength properties may break and tear. To enable the selection of materials with proper response against the strain that occur during crack movement, this study proposes and demonstrates a new evaluation method for determining and comparing strain concentration of waterproofing materials under the effect of concrete crack movement. For the proposed testing method and demonstration, three common types of waterproofing material types were selected for testing, poly-urethane coating (PUC), self-adhesive asphalt sheet (SAS) and composite asphalt sheet (CAS). Respective materials are installed with strain gauges and applied onto a specimen with a separated joint that undergoes concrete crack movement simulation. Each specimen types are subject to repeated movement cycles, whereby strain occurring directly above the moving joint is measured and compared with the strain occurring at the localized sections (comparison ratio which is hereafter referred to as strain concentration ratio). Specimens are tested under four separate movement length conditions, 1.5 mm, 3.0 mm, 4.5 mm and 6.0 mm, and the results are compared accordingly. Experimental results show that materials with strain concentration ratio from highest to lowest are as follows: PUC, SAS and CAS.

## 1. Introduction

The purpose of this study is to propose a new waterproofing material evaluation method that enables the selection of material types that are able to respond to varying degrees of concrete crack movement by measurement and comparison of strain occurring at the crack joint and localized sections of the installed waterproofing layer. Concrete structures can be affected by various forms of complex environments, due to settlement, thermal variation, vibration related load, etc., and concrete cracks eventually occur [1]. Despite this, contractors often opt to use affordable waterproofing materials that are not always suitable to withstand the structural degradation and environmental conditions, which can lead to waterproofing failure and leakage. A common failure among waterproofing materials is cohesive and adhesive failure at the adhesion interface and this is particularly apparent with materials with high tensile strength but low elongation (brittle materials such as cementitious or crystalline waterproofing materials) [2].

Existing studies indicate that circumstances with infrastructures, particularly that of railway structures, accord with this problem. Nielson et al. [2] conducted a research on estimated life cycle cost for railway bridge maintenance, where a focus on using a proper waterproofing system to ensure the long-term durability of bridge structures against cracks caused by corrosion is required. Zhang provided an overview of common problems found in bridge decks due to leaks and lack of adequate waterproofing and proposed selection criteria for the waterproofing layer method to avoid these common problems [3]. Xu reviewed various factors affecting the interfacial adhesion of the waterproofing material to the concrete surface, including surface roughness, material properties and thickness and temperature [4]. He suggested that the adhesive strength initially increases and decreases as the film thickness increases and the same mechanism applies to surface roughness. For railway structure bridges, securing long-term safety due to an increase in running speed and lengthening of the top plate has become an important task and reinforcement of waterproofing performance to ensure continuous durability and safety has become an important research subject. Dammyr et al. discussed Norwegian waterproofing and compared other general European waterproofing concepts for securing high-performance waterproofing in railway tunnels; in the case of bridge deck structures, manuals for waterproofing of concrete bridge decks exist in the United States, Scotland, Canada and the United Kingdom [5]. Su and Bloodworth also proposed a numerical analysis method for spray-in-water waterproofing in tunnels that can simulate complex actions and provide a recommended design for spray-applied waterproofing; complex actions, such as tension, compression and shear forces, were studied regarding how they affect the integrity of waterproofing adhesion [6]. Nozomu Taniguchi examined the mechanisms of how waterproofing protects the concrete substrate deck of the railway bridges based upon a comparison of concrete fracture rate difference between different types of waterproofing materials and the requirement of waterproofing for protecting the railway bridge against train loads [7]. In the field of nano materials, Kai Yin et al also provided articles on the importance of hydrophobic–hydrophilic surfacing for fog collection and PTFE film application for self-cleaning function for passive-cooling materials [8,9].

Existing test standards (such as ASTM, BS EN, KS, GB, JS, etc.) include environmental degradation and joint movement around the structure and there is currently no known method to evaluate waterproofing material properties based on strain measurement during concrete crack movement. A review of current standard waterproofing material testing methods showed that waterproofing materials are only assessed based on their physical characteristics and properties [10,11,12]. A new evaluation that is able to assess the waterproofing materials that are under the effect of concentrated tension at the point of movement is needed, as waterproofing materials of different material properties, elastic modulus and bonding characteristic respond differently, in that the strain on the waterproofing material above the movement interface is different at the joint and localized sections (where higher strain concentration at the point of movement indicates higher probability of fracture or defect occurrence on the waterproofing material).

In this regard, this study proposes a new evaluation method and criteria for waterproofing materials that assesses and grades the capacity of the material to respond to crack and/or concrete crack movement by measurement of strain concentration ratio. Waterproofing materials were installed onto specialized specimen designed to simulate movement and the strain occurring on the waterproofing materials was measured at the movement joint and upper and lower sections, respectively, using strain gauges. Strain occurring at the movement joint (hereby called center section) was compared to the strain occurring at upper and lower sections of the specimen, whereby the strain concentration ratio can be derived for each waterproofing material type. Based on these results, the ratios can be used as a form of grading system for the tested waterproofing materials and compared for performance evaluation. The study demonstrates this process by a mock testing using three common types of waterproofing materials that comply to the standard specification (poly-urethane coating (PUC), self-adhesive asphalt sheet (SAS) and composite asphalt sheet (CAS)).

## 2. Theoretical Discussion

As was discussed in the introduction, cyclic movements of concrete cracks and/or joints can cause waterproofing material to be subject to concentrated strain at the adhesion interface over time [13]. For concrete bridge structures, the design guide indicates that there is a strong probability of waterproofing failure occurring across the movement joint at the bridge deck sections [14]. In this case, the effect of zero-span tensile stress (localization of strain around the moving joint/crack on the adhered waterproofing material) can affect the adhesive and cohesive bond of the waterproofing material.

As cracks in the concrete substrate are eventual and inevitable over time, it is pertinent, for waterproofing materials, to have sufficient response properties such that remedial actions are not necessary even under a high degree of concrete crack movement. Many types of existing waterproofing material products are already more than capable of withstanding high movement width and maintain a watertight protection against hydrostatic pressure. The main concern is that these materials are often more costly than the average types of waterproofing products and the appropriate methodology of evaluation that aptly highlights this particular property of the waterproofing materials is not practiced. The above review of existing studies indicates that research on standard waterproofing/durability/combined deterioration/longevity/behavior response in structures is important, but these studies also highlight problems where waterproofing material suitable for the conditions and deterioration environmental conditions is not often used. In an effort to remediate this situation, this study proposes a new evaluation method that can be used to clearly outline the difference in the response performance against concrete crack movement.

### 2.1. Importance on the Measurement of Strain on Waterproofing Materials during Concrete Crack Movement

Strain (deformation) of a material can generate internal stress and it is crucial to understand the purpose and design of a concrete structure to estimate the expected amount of physical movement that can potentially occur, so as to prepare a suitable waterproofing type. For cementitious waterproofing types that are inherently brittle in nature, strain is expected to be higher at the point of movement interface (due to concrete substrate crack or joint movement); therefore, high probability of failure is expected, if this movement is too high and frequent. On the other hand, coating types of waterproofing are expected to respond better to crack movement, due to their intrinsic elastic properties, meaning strain is less concentrated at the crack or joint movement interface. For sheet types of waterproofing, the rubbery sealant layer at the adhesion surface allows even higher response against concentrated strain at the crack or joint movement than the previous two types; however, with sheet types, it is more difficult to secure high adhesion and they are more costly. Refer to below Figure 1 for the illustrated concept. 

In materials with a flexible layer with high elongation at the interface of the waterproofing material and the concrete surface, the transfer mechanism of strain is different, whereas brittle materials are expected to have high strain concentration. However, this is only by theoretical estimation, as there has not yet been any attempts at evaluating different types of waterproofing materials empirically through strain measurements. By simulating this movement, we can test this theory. Strain measurement can be used to compare how crack movement affect the waterproofing layer by comparing varying degrees of strain occurring directly above the crack movement interface and the surrounding sections. Strain within the waterproofing material can be measured using Equation (1); the deformation (change in length) of the waterproofing layer in conjunction to the moving crack or joint can be expressed by the following:(1)$$\varepsilon =\frac{\Delta L}{L}$$
where ΔL = change of length (mm), *L* = initial length (mm) and ε = strain.

### 2.2. Explanation of Strain Concentration Ratio Limited to Waterproofing Material Property

Waterproofing materials with movable cracks present underneath are subjected to high tension, a phenomenon which is referred to as ‘strain concentration’ hereafter, for the scope of this study. Strain concentrations occur when there are irregularities in the geometry or material of a structural component that cause an interruption of the flow of strain [15]. When subjected to cycles of this type of concentrated strain at the crack/joint adhesion interface, defects and fractures can occur on the adhered waterproofing material layer, affecting the adhesive bond between the waterproofing material and concrete surface, as well as and the cohesive bond within the material itself [16]. This concept is illustrated in Figure 2 below, where *L_w_* represents the change in the length of the waterproofing layer, while *l_c_* is the length of movement of the concrete (either a crack or joint, commonly occurring due to the freeze–thaw effect or vibration, depending on the structure and surrounding environment). While the change in the *l_c_* may be predictable, depending on the size of the crack/joint and the environmental factors involved, the change in the length of the waterproofing layer (*L_w_*) depend on the type of the waterproofing layer. If the concerned waterproofing layer is comprised of a cementitious material, a constant length may be applied to both *L_w_* and *l_c_*, whereas, for example, for certain compositely structured waterproofing sheets, there is barely any change to *L_w_*.

For the sake of simplification, the comparison between the strain on the waterproofing material directly above the movement interface and that of the surrounding regions will hereby be referred to as ‘strain concentration ratio’ for waterproofing materials. Strain generated at these interfaces would differ based on the different properties of the material types and can be used as a factor of comparison for evaluation.

## 3. Experimental Regime

### 3.1. Selected Waterproofing Materials for Testing

For the evaluation demonstration, three types of waterproofing systems were selected, (1) polyurethane spray coating (PUC), in a liquid applied material system, (2) self-adhesive asphalt sheet (SAS), in an asphalt sheet system, and (3) composite asphalt sheet (CAS), in an asphalt sheet system. Based on the international setting research discussed in the previous sections, product types with the most frequent usage in recent construction history were surveyed [17]. Among the 3 types, materials with the clearest variances were chosen intentionally, as the demonstration of the test method is intended to illustrate the difference in the performance of the different classification of the materials. Refer to Table 1, Table 2 and Table 3 for details of the material specifications.

### 3.2. Specimen Preparation for Installation of Waterproofing Systems for Testing

In order to design and demonstrate this evaluation method by strain concentration ratio comparison, a specimen was constructed such that the three types of waterproofing materials could be installed over a set of concrete/mortar substrate slabs with a movable joint that represented a crack on a concrete substrate. The base component for the concrete specimen was comprised of upper and lower concrete substrate parts installed with threaded conduit that were used to connect the specimen to a universal testing machine apparatus. Threaded conduits were placed in their corresponding substrate parts which were used for connection to the testing device. The test specimen was comprised of upper and bottom cylindrical mortar slab parts. The two parts were placed together to form a separation gap at the interface. This gap represented a concrete joint range (of width). Refer to Figure 3 for illustration.

The installation of waterproofing materials was conducted by representatives of the manufacturers, to ensure workmanship compliant with the specifications. For each waterproofing system, 5 specimens were prepared. The waterproofing material was cut into a 650 by 150 mm rectangular piece. The material was installed on the mortar slabs placed together with the short dimension applied perpendicular to the joint gap. When applying the waterproofing material sheets, an overlap joint with a minimum width of 50 mm was made. When the waterproofing material was adhered over the joint, a 50 mm of exposed area from the slab edge to the waterproofing material formed. The installation was conducted in a laboratory setting with ambient conditions (temperature of 20 ± 3 °C, relative humidity of 60 ± 5%). Specimens with complete installation were set to rest in a curing room accordingly to the manufacturer’s specifications. The strain gauge was attached to the surface of the waterproofing layer coated with the movement test specimen (9 points, labelled A–I) and the strain value was measured to compare the difference of strain occurring throughout the different sections relative to the center section (strain gauge points E, B and H). Refer to Figure 4 below for illustration of the concept.

### 3.3. Movement Simulation Concept for Strain Concentration Ratio Measurement

The specimen was installed onto the crack/joint movement simulator testing machine (universal testing machine built in), where the threaded conduit of the upper substrate part was anchored, whereby the tensile action, by pulling the upper substrate part, induced tensile force at the movement interface of the waterproofing material. Refer to Figure 5 below for an illustration of the testing apparatus, specimen installed with strain gauges and data acquisition system apparatus (currently not standardized).

The movement width for this evaluation demonstration was set at 4 intervals, 1.5, 3.0, 4.5 and 6.0 mm (within tolerance ± 0.2 mm), and the movement speed was set to 50 mm/min. Cycles were repeated for up to 100 times and the data for the last 15 cycles were selected for measurement and analysis. After the upper substrate was pulled up once, the load-movement simulator set the specimen back to the original position and the cycle repeated. Refer to Figure 6 for details and illustration.

## 4. Results and Discussion of Stress–Strain Evaluation Effect on Waterproofing Layer

### 4.1. Strain Measurement Results 

Strain measurements at the four movement intervals showed that the derived strain was different from the strain measured at the upper parts (strain gauge points A, D and G) and bottom parts (strain gauge points C, F and I) compared to the center points (B, E and H). Refer to Table 4 for a sample measurement result table where the averaged maximum strain values obtained from measurement (peak of last 15 cycles) are shown and Figure 7, Figure 8 and Figure 9 for the example results of the strain measurements.

Comparing the strain at the upper and bottom sections respectively, the increase in the strain difference at the center section (correlative to the increase of movement width) indicates that strain is certainly higher at the center, with low values of strain being obtained in the case of PUC and SAS specimens. In the case of CAS, however, it is expected that, due to the gel compound, strain is distinctly less high than the values obtained at upper and bottom sections, as evidenced by the more intermittent peaks shown in Figure 9.

### 4.2. Strain Comparison of Waterproofing Materials Based on Averaged Maximum (Peak) Strain Values

Experimental results for the averaged peak strain at the center sections, upper sections and bottom sections showed high strain at the center section. To calculate the strain concentration ratio for the respective waterproofing material types, the average strain occurring at the center sections was calculated (using Equation (1) and the modulus of elasticity provided in Table 1, Table 2 and Table 3, for the respective material types). Refer to Table 5 below for averaged peak strain obtained for the center sections for example details (also conducted for upper and bottom sections).

### 4.3. Waterproofing Material Strain Concentration Ratio Derivation and Comparison

The averaged strain for each movement range (from 1.5 mm to 6.0 mm) at the center section was set as the base value for comparison and the difference of strain derived at the averaged strain at the upper and bottom section was derived. The results are shown in Figure 10, Figure 11 and Figure 12, where the average strain (among five specimens and three gauge points for the respective upper and lower sections) are differenced relative to the center base strain value. 

The differences between the base (center strain value) and the upper and lower sections are added together to derive an overall strain concentration ratio (values of upper and bottom concentration ratios are added together). The results of this strain difference for the respective waterproofing material types, in accordance with the movement width ranges, were calculated into a format of strain concentration ratio and are provided in Table 6 below. Based on these results, a linear regression analysis of the changing strain concentration ratio relative to the increasing movement width for each waterproofing material types was conducted and compared for easier visual comparison (provided in Figure 13.)

## 5. Conclusions

The experimental results show that, for the waterproofing materials considered, strain concentration ratios increase from 1.5 mm to 6.0 mm at difference rates. Due to the high elongation and modulus of elasticity nature of the gel material under the film of CAS waterproofing material, a lower range of strain concentration was expected and evidenced than those of PUC and SAS. Furthermore, changes to the strain concentration ratio relative to the increasing movement width (Figure 13) showed that while the CAS type of waterproofing material maintains similar strain concentration ratio throughout all of the width conditions, PUC has the highest rate of increase, followed by the SAS type of material. The study results are based on a demonstration of this newly proposed evaluation procedure, using material types of clearly distinct performance and characteristics. Future application of this evaluation method need to be conducted for a wider variety of materials used in different nations with a wider range of conditioning. However, for the purpose of this study, the evaluation method and the results of this study were able to clearly demonstrate that waterproofing materials have different degrees of strain concentration response to concrete crack movement. 

## Figures and Tables

**Figure 1 materials-14-04429-f001:**
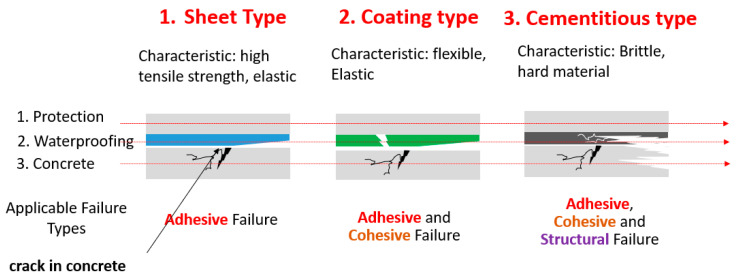
Tensile strain concentration concept (hypothetical) for different types of waterproofing materials.

**Figure 2 materials-14-04429-f002:**
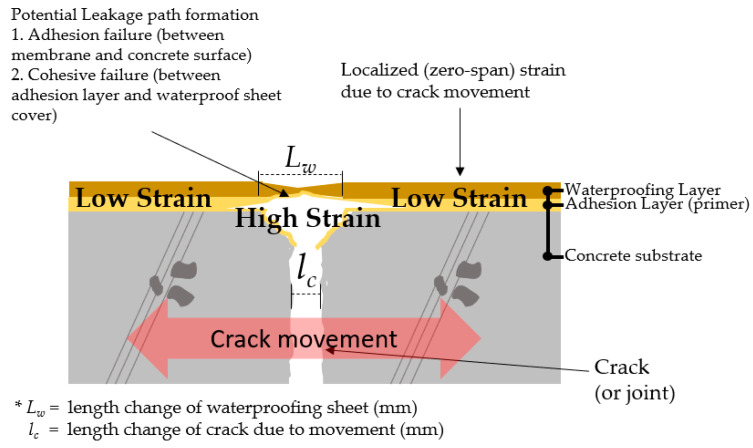
Strain concentration directly above the moving crack on waterproofing material illustrated.

**Figure 3 materials-14-04429-f003:**
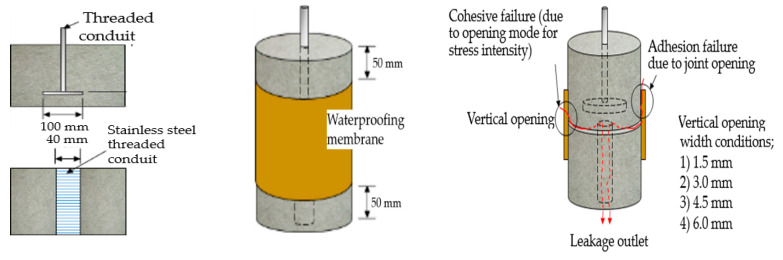
Movement simulation specimen for strain concentration ratio evaluation.

**Figure 4 materials-14-04429-f004:**
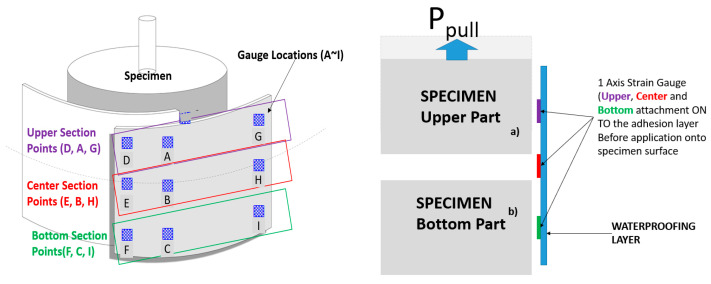
Strain gauge placement and measurement concept.

**Figure 5 materials-14-04429-f005:**
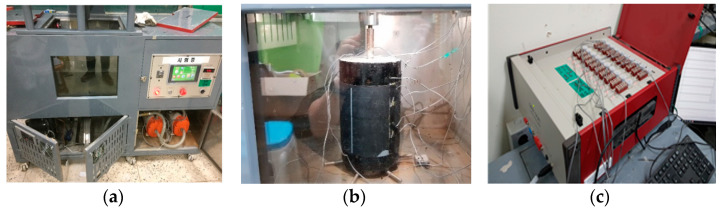
Testing apparatus illustrated: (**a**) crack/joint movement simulator testing machine (UTM) (Construction Technology Research Center, Nowon-gu, Seoul, Korea), (**b**) specimen installed with strain gauges and (**c**) data acquisition system apparatus.

**Figure 6 materials-14-04429-f006:**
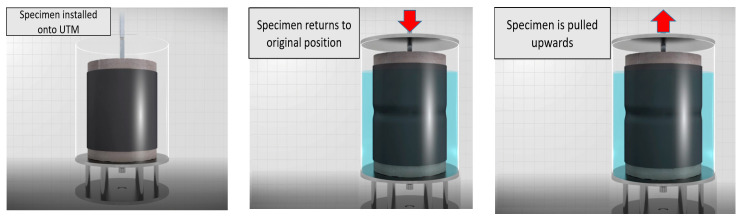
Strain concentration ratio evaluation method by concrete joint movement simulation.

**Figure 7 materials-14-04429-f007:**
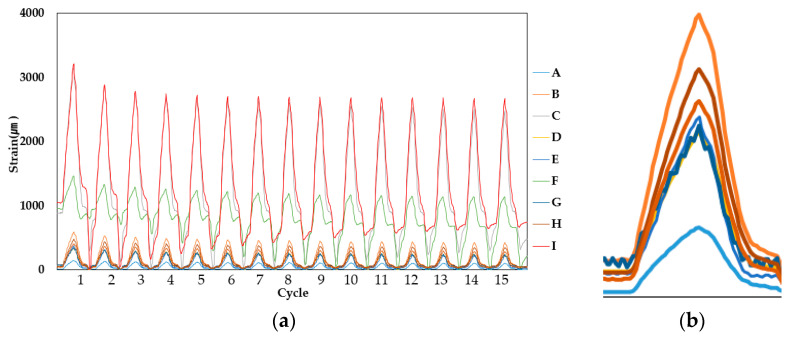
PUC (Specimen 1, 1.5 mm specimen sample) strain measurement result: (**a**) overall result and (**b**) single peak results of upper and lower section strain, magnified.

**Figure 8 materials-14-04429-f008:**
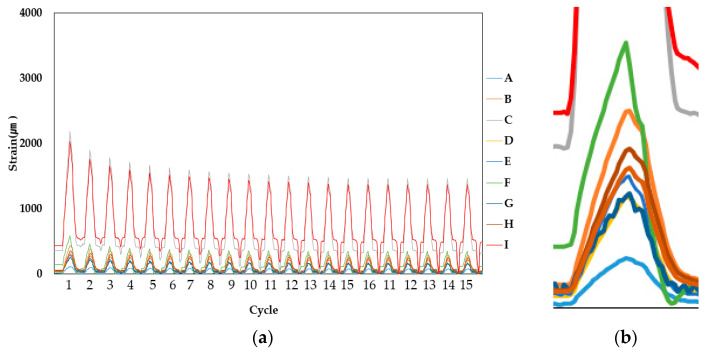
SAS (Specimen 1, 1.5 mm specimen sample) strain measurement result: (**a**) overall result and (**b**) single peak result of upper and lower section strain, magnified.

**Figure 9 materials-14-04429-f009:**
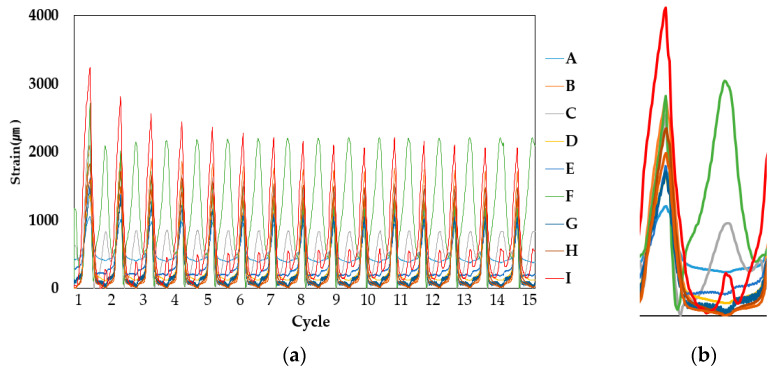
CAS (Specimen 1, 1.5 mm specimen sample) strain measurement result: (**a**) overall result and (**b**) single peak result, magnified.

**Figure 10 materials-14-04429-f010:**
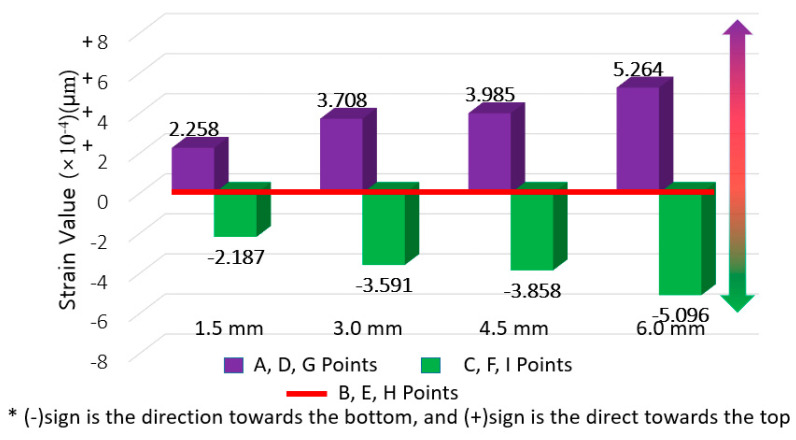
PUC strain difference results (for deriving strain concentration ratio).

**Figure 11 materials-14-04429-f011:**
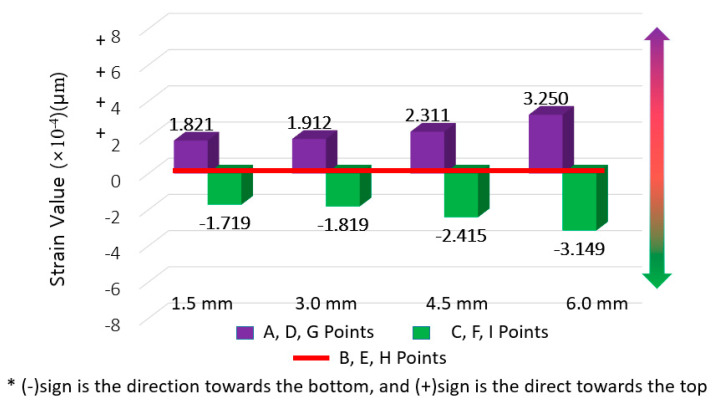
SAS strain difference results (for deriving strain concentration ratio).

**Figure 12 materials-14-04429-f012:**
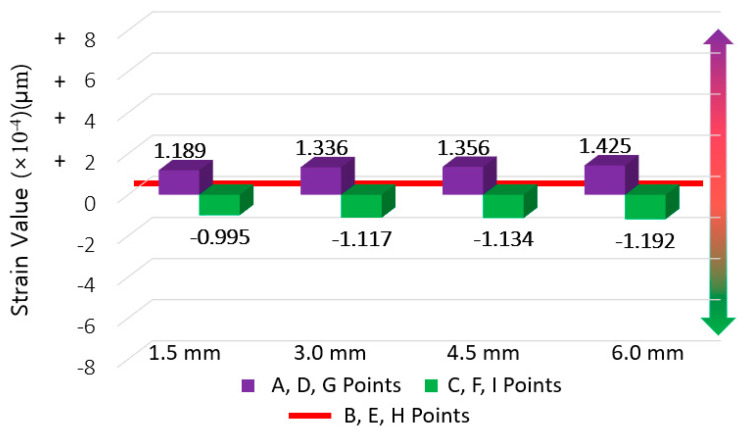
CAS strain difference results (for deriving strain concentration ratio).

**Figure 13 materials-14-04429-f013:**
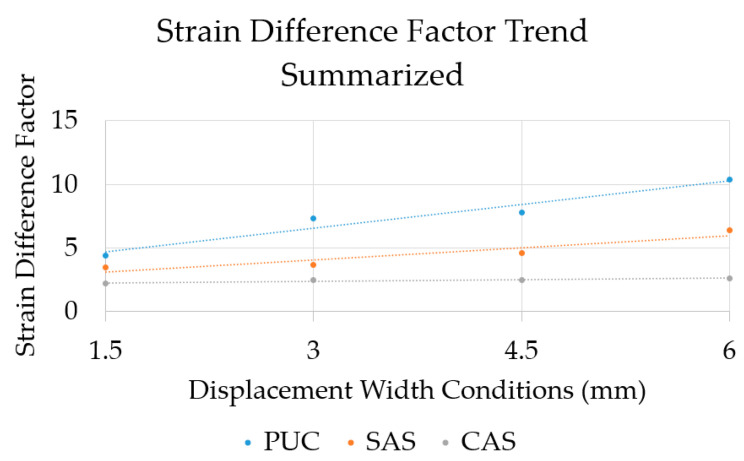
Strain concentration ratio linear regression analysis in accordance with movement width.

**Table 1 materials-14-04429-t001:** Poly-urethane coating (PUC) specification.

Items					Standard
Tensile performance	Tensile strength		N/mm^2^		More than 2.5
	Elongation		%		More than 450
Tear resistance			N/mm		More than 14.7
Heated contraction ratio			Contraction	%	More than −4, Less than 1
Tensile performance after deterioration	Tensile strength ratio	%	Heating		More than 80 Less than 150
			Accelerated aging		More than 80 Less than 150
			Alkali		More than 60 Less than 150
			Acid		More than 80 Less than 150
	Elongation	%	Heating		More than 400
			Accelerated aging		
			Alkali		
			Acid		
Adhesion strength (peel-out)			Untreated	N/mm^2^	More than 0.7
			Thermal variation		More than 0.5
					No crack or defect on the waterproofing surface
Modulus of elasticity			14.5 GPa		

**Table 2 materials-14-04429-t002:** Self-adhesive asphalt sheet (SAS) specification.

Items					Standards
Tensile strength		Tensile strength,	N/mm	Length	More than 3.0
				Width	
		Elongation	%	Length	More than 200
				Width	
Tear strength			N	Length	More than 25
				Width	
Thermal stress	60 °C	Tensile strength,	N/mm	Length	More than 2.0
				Width	
		Elongation after break	%	Length	More than 150
				Width	
Thermal stress	−20 °C	Tensile strength,	N/mm	Length	More than 5.0
				Width	
		Elongation after break	%	Length	More than 50
				Width	
Adhesion stability		Permeability			No hydrostatic penetration
		Peel off resistance		N/mm	More than 1.5
Adhesion strength (peel-out)				N/mm	More than 1.5
Modulus of elasticity				0.4 GPa	

**Table 3 materials-14-04429-t003:** Composite asphalt sheet (CAS) specification.

Items					Standard
Tensile performance	Tensile strength		N/mm^2^		More than 2.5
	Elongation		%		More than 450
Tear resistance			N/mm		More than 14.7
Heated contraction ratio			Contraction	%	More than −4, Less than 1
Tensile performance after deterioration	Tensile strength ratio	%	Heating		More than 80 Less than 150
			Accelerated aging		More than 80 Less than 150
			Alkali		More than 60 Less than 150
			Acid		More than 80 Less than 150
	Elongation	%	Heating		More than 400
			Accelerated aging		
			Alkali		
			Acid		
Adhesion strength (peel-out)			Untreated	N/mm^2^	More than 0.7
			Thermal ariation		More than 0.5
					No crack or defect on the waterproofing surface
Modulus of elasticity (gel)			1.14 GPa		

**Table 4 materials-14-04429-t004:** Sample strain measurement results for each strain gauge point (averaged peaks).

Points	Strain Max Value (×10^−4^) (μm) per Specimen				
	1	2	3	4	5
A (center)	4.474	5.014	4.082	5.034	3.867
B (upper)	0.493	0.553	0.450	0.556	0.427
C (lower)	0.537	0.602	0.490	0.605	0.464
D (center)	2.090	2.344	1.907	2.352	1.807
E (upper)	0.493	0.553	0.450	0.556	0.427
F (lower)	0.680	0.764	0.622	0.766	0.589
G (center)	4.596	5.153	4.193	5.173	3.973
H (upper)	0.511	0.573	0.467	0.576	0.441
I (lower)	0.587	0.658	0.536	0.660	0.507

**Table 5 materials-14-04429-t005:** Strain measurement results for each strain gauge points (averaged peaks).

Specimens # (Strain (ε) at Center Section)	PUC				SAS				CAS			
Width (mm)	1.5	3.0	4.5	6.0	1.5	3.0	4.5	6.0	1.5	3.0	4.5	6.0
1	1.41	3.72	6.05	7.45	2.93	3.35	3.46	3.65	1.31	1.84	2.35	3.46
2	1.58	4.17	6.78	8.35	3.29	3.75	3.88	4.26	1.33	1.88	2.38	3.70
3	1.28	3.39	5.52	6.80	2.68	3.05	3.16	3.33	1.27	1.81	2.31	3.33
4	1.58	4.19	6.81	8.38	3.30	3.77	3.90	4.31	1.32	1.85	2.36	3.72
5	1.22	3.22	5.23	6.44	2.54	2.89	2.99	3.16	1.29	1.83	2.34	3.26
Average peak strain	1.41	3.22	5.23	6.44	2.95	3.36	3.48	3.74	1.31	1.84	2.35	3.55

**Table 6 materials-14-04429-t006:** Strain concentration ratio comparison of the waterproofing materials.

Movement Width Range (mm)	Strain Concentration Ratio (Relative to the Center Section)		
	PUC	SAS	CAS
1.5	4.4	3.5	2.2
3.0	7.3	3.7	2.5
4.5	7.8	4.6	2.5
6.0	10.4	6.4	2.6

## Data Availability

Not applicable.

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
