# Peer review of "Strain Concentration Ratio Analysis of Different Waterproofing Materials during Concrete Crack Movement"

_materials, 2021, doi:10.3390/ma14164429_

Round 1

Reviewer 1 Report

Kyu-hwan Oh et al. investigated a novel evaluation method for determining the stress distribution resistance performance of waterproofing materials during concrete displacement. Moreover, three kinds of waterproofing material were studied. It is novel and a topic of interest to the researchers in the related areas. And it could publication after the following minor problems addressed.

  1. The image of practical experimental test equipment could be exhibited in the manuscript.
  2. The data lines in Fig.7-9 could not see clearly. And the corresponding local magnification images could be given.
  3. For the studies of waterproofing materials and applications, the authors may refer these papers:1. Nano Letters, 2021, 21, 4209-4216; 2. Nanoscale, 2017, 9 (38), 14620-14626;
  4. For more perfection, several language mistakes could be revised.

Author Response

The authors of article materials–1307532 would like to extend their sincerest gratitude to the reviewers for taking time out of their busy schedule to review our paper. Much thanks to your efforts, comments and contribution, we hope that the paper has improved in quality and clarity. The below lists the comments provided by the reviewers and the respective response and details as to how the comments were address and applied in the revised article.

Reviewer Introduction

Kyu-hwan Oh et al. investigated a novel evaluation method for determining the stress distribution resistance performance of waterproofing materials during concrete displacement. Moreover, three kinds of waterproofing material were studied. It is novel and a topic of interest to the researchers in the related areas. And it could publication after the following minor problems addressed.

Author Comment

The authors are once again very grateful for the reviewer’s kind and encouraging words for the article.

Reviewer Comment 1

The image of practical experimental test equipment could be exhibited in the manuscript.

Author Response 1

Images of practical experimental test equipment, specimen, and supporting apparatus have been included in the manuscript (Please refer to now Figure 5. Line 245 of the revised manuscript)

Reviewer Comment 2

The data lines in Fig.7-9 could not see clearly. And the corresponding local magnification images could be given.

Author Response 2

The data lines for Fig.7-9 have been revised such that local magnification of one peak section are provided at the side (please refer to the revised version of the manuscript, Figure 7-9)

Reviewer Comment 3

For the studies of waterproofing materials and applications, the authors may refer these papers:1. Nano Letters, 2021, 21, 4209-4216; 2. Nanoscale, 2017, 9 (38), 14620-14626;

Author Response 3

References to the above two mentioned articles have been included in the introduction section of the manuscript (please refer to lines 66 to 68 of the revised manuscript)

Reviewer Comment 4

For more perfection, several language mistakes could be revised.

Author Response 4

Manuscript has been revised such that language and technical mistakes were corrected

The authors would like to express their thanks once again for the reviewer’s time and valuable feed back on this manuscript.

Reviewer 2 Report

The article was a disappointment for me. The idea itself and the research carried out seem interesting. The work, however, is very imperfect.

1. The title itself is disturbing. It can only be understood after reading a few pages. The abstract and the introduction alone are not enough. The authors use phrases that are imprecise or not commonly used. It is a mistake to use the term "stress dispersion analysis", because until you define what the authors mean, this term does not say anything. "Concrete displacement" is also unclear because concrete is the designation of a material, not an element or structure that could move.

2. The abstract needs to be completely redone. The expressions are imprecise and / or inappropriate. I have no idea what stress distribution resistance performance and displacement stress mean.

3. In the introduction, the article was properly supported by citations. In the introduction, the editorial and language side should be corrected.

4. Chapter 2 (Theoretical discussion) makes me think that the world is chaos. Figure 1 is unclear. What does A: mean? What is displacement length? From a mechanical point of view, displacement is a displacement - measured, for example, in meters. So why use the term displacement length? This definition of the stress intensity factor is given, but it is not explained what f_ij is, what is agrument theta, what is r, etc. It looks as if the figure was redrawn from another publication.
Figures 2 and 3 are unclear.
The yield strength is a feature of a material, not a method. It seems to me that I know what the authors mean, but you cannot be so imprecise.

5. Equation (1) together with the description of the variables is in my opinion incorrect. I omit that in the equation y is a lower case letter and in the description Y is a capital letter. And who is to know if it's the same or not? What is it anyway? Not shown anywhere. Why R refers to the metallic material? Add units.

6. Section4 is difficult to understand at the moment, but that is likely to change if the earlier sections change. The conclusions are too general. The specific effects resulting from the conducted research should be indicated. 

7. Linguistically, the work is poor, mainly due to imprecise (and / or incorrect) sentences.

Author Response

The authors of article materials–1307532 would like to extend their sincerest gratitude to the reviewers for taking time out of their busy schedule to review our paper. Much thanks to your efforts, comments and contribution, we hope that the paper has improved in quality and clarity. The below lists the comments provided by the reviewers and the respective response and details as to how the comments were address and applied in the revised article.

Reviewer Introduction

The article was a disappointment for me. The idea itself and the research carried out seem interesting. The work, however, is very imperfect.

Reviewer Comment 1

The title itself is disturbing. It can only be understood after reading a few pages. The abstract and the introduction alone are not enough. The authors use phrases that are imprecise or not commonly used. It is a mistake to use the term "stress dispersion analysis", because until you define what the authors mean, this term does not say anything. "Concrete displacement" is also unclear because concrete is the designation of a material, not an element or structure that could move.

Author Response 1

The title has been revised to; “Strain Concentration Ratio Analysis of Different Waterproofing Materials during Concrete Crack Movement” in order to remove confusion. As adequately pointed out by the reviewer, this “property/characteristic” outlined by the topic of this paper is rather new in the field of evaluating waterproofing materials. The terms and expressions were originally intended to align with other previously published papers related to this theme to maintain consistency, but as pointed out by the reviewer, revisions have been made. The terms strain concentration ratio has been used instead of stress dispersion/distribution to focus on the underlined problem of waterproofing materials not having sufficient response property during concrete crack movement (replacing concrete displacement) whereby defects leading to leakage path formation due to fracturing and adhesion failure. Please refer to the revised manuscript for details

Reviewer Comment 2

The abstract needs to be completely redone. The expressions are imprecise and / or inappropriate. I have no idea what stress distribution resistance performance and displacement stress mean.

Author Response 2

The abstract has been rewritten using the new terminologies, and the goal of the research outlined in the paper has been made more clear as well. Please refer to the revised abstract (Lines 11 to 26) for details.

Reviewer Comment 3

 In the introduction, the article was properly supported by citations. In the introduction, the editorial and language side should be corrected.

Author Response 3

Citation supports now properly accord with the referred studies in the introduction, and editorial revisions have been made. Also, most of the introduction has been rewritten for clarity purposes. Please refer to the revised manuscript lines 32 – 79 for details

Reviewer Comment 4

 Chapter 2 (Theoretical discussion) makes me think that the world is chaos. Figure 1 is unclear. What does A: mean? What is displacement length? From a mechanical point of view, displacement is a displacement - measured, for example, in meters. So why use the term displacement length? This definition of the stress intensity factor is given, but it is not explained what f_ij is, what is agrument theta, what is r, etc. It looks as if the figure was redrawn from another publication. Figures 2 and 3 are unclear. The yield strength is a feature of a material, not a method. It seems to me that I know what the authors mean, but you cannot be so imprecise.

Author Response 4

Section 2 has been revised and completely rewritten to clarify the purpose of the paper. Figure 2 has been removed to avoid confusion, and Figure 3 (now Figure 1 in the revised manuscript) have been revised to accord with the revisions made in the manuscript. Please refer to the revised manuscript Lines 96-184 for details.

Reviewer Comment 5

 Equation (1) together with the description of the variables is in my opinion incorrect. I omit that in the equation y is a lower case letter and in the description Y is a capital letter. And who is to know if it's the same or not? What is it anyway? Not shown anywhere. Why R refers to the metallic material? Add units.

Author Response 5

As the concept has been reoriented towards a focus on measuring and comparing strain on the waterproofing material, the relevant equations have been simplified and changed throughout the paper (please refer to the revised Equation (1) in line 179 and Equation (2) in Line 258 for details). Explanations for symbol R has been outlined more clearly (lines 255-256 in the revised manuscript)

Reviewer Comment 6

Section 4 is difficult to understand at the moment, but that is likely to change if the earlier sections change. The conclusions are too general. The specific effects resulting from the conducted research should be indicated. 

Author Response 6

Section 4 has been revised accordingly to the changes made in the previous sections. Please refer to the revised section 4 for details.

Reviewer Comment 7

Linguistically, the work is poor, mainly due to imprecise (and / or incorrect) sentences.

Author Response 7

Manuscript has been revised such that language and technical mistakes were corrected

The authors would like to express their thanks once again for the reviewer’s time and valuable feed back on this manuscript.

Round 2

Reviewer 2 Report

1. Subsectio 2.1 should be reorganized and clarified:
a) line 162 "Any strain (deformation) of a material generates an
internal elastic stress" - in my opinion it does not have to be elastic
b) line 163-164 "If stress exceeds the yield strength of the material,
this can result in permanent deformation". This is also a risky
sentence. The authors (incorrectly in my opinion) assume certain
specific behavior of the material. Not all materials have yield
strengths, and not all materials have permanent deformations.
c) The authors write about the following types of materials:
cementitious, coating type, composite type. But this is a confusion,
because the first name indicates the type of material, and the others
indicate the structure rather than the material. Besides, authors mix
even more using the terms "brittle type" (type of behaviour), "single
ply coating" (more detailed than coating type?), sheet or film (?) ...
Everything should be better organized together with Fig. 1

2. Line 197-198: improve syntax.

3. Equation (1) is a classic, but when I read the authors' explanations,
I don't know what they mean. The increment of L is the the waterproofing
material elongation? How is it measured? If a strain gauge, what is its
base length L? If L denotes the moving crack or joint, I don't
understand any of it anymore.

4. Line 223: rather "are subjected" not "are subject". "to higher
tension due to stress intensity factor" - no factor will change the
strain, stress or deformation.

5. Figure 2. The explanation of Lw and lc is debatable. If we assume a
certain initial state, then extending the connection by a certain value
of x should change both lengths by x (simplified). If we use more
advanced dependencies, they should be described in detail.

6. I still don't know what the "strain concentration ratio" is. There is
no definition in the mathematical sense.

7. Figure 4. How far from the connection are the strain gauges? In
figure 4 (on the right) looks as if the strain gauges were not mounted
on the adhesive layer, but directly on the specimen.

8. What is the point of giving the equation (2)? And if I understand
correctly - giving resistance units incorrectly? If we are talking about
electresistant strain gauges, the resistance will be in ohms ... (?) Or
I don't know what the authors mean.

9. The article presents the results of the experiment. However, if a
definition of "strain concentration ratio" does not appear, they will be
difficult to understand.

Author Response

The authors would like to express their sincerest gratitude to the reviewer for once again taking time to review the manuscript materials-1307532. Below are the responses to the comments/points of revision provided by the reviewer;

  1. Subsection 2.1 should be reorganized and clarified

a) line 162 "Any strain (deformation) of a material generates an

internal elastic stress" - in my opinion it does not have to be elastic

Line 162 has been revised accordingly

b) line 163-164 "If stress exceeds the yield strength of the material,

this can result in permanent deformation". This is also a risky

sentence. The authors (incorrectly in my opinion) assume certain

specific behavior of the material. Not all materials have yield

strengths, and not all materials have permanent deformations.

As is mentioned in the paper, only some waterproofing materials undergo plastic/permanent deformation. The point of this section is to illustrate that some waterproofing materials how low yield strength, susceptible to change after repeated cycles of stress or small concrete movement when the concrete substrate cracks, while others have crack bridging properties. It was never the intention of the author to claim that all materials undergo permanent deformation.

c) The authors write about the following types of materials:

cementitious, coating type, composite type. But this is a confusion,

because the first name indicates the type of material, and the others

indicate the structure rather than the material. Besides, authors mix

even more using the terms "brittle type" (type of behaviour), "single

ply coating" (more detailed than coating type?), sheet or film (?) ...

Everything should be better organized together with Fig. 1

The explanation on the different types of waterproofing have been clarified such that aforementioned (coating, sheet, and cementitous) waterproofing concerned in this manuscript is dealt with as types, not materials for the theoretical discussion. This is intended as a generalization of very common types of waterproofing. Sheet and coating types can be classified into double-ply or single-ply types, of which in this case explains that single-ply types in particular are more vulnerable to defect occurrence due to repeated cycles of concrete movement. Figure 1 has been replaced with a simpler image for better illustration and explanation on the difference between the waterproofing types. Explanation has been supplemented in Lines 122-131 in the revised manuscript.

  1. Line 197-198: improve syntax.

Lines 197 to 198 has been revised;

“Strain measurement can be used to compare how crack movement affect waterproofing layer by comparing varying degrees of strain occurring directly above the crack movement interface and the surrounding sections.”

  1. Equation (1) is a classic, but when I read the authors' explanations,

I don't know what they mean. The increment of L is the waterproofing

material elongation? How is it measured? If a strain gauge, what is its

base length L? If L denotes the moving crack or joint, I don't

understand any of it anymore.

Equation 1 is simply intended as a reference for readers. It is measured in percentages, and it defines the amount of which a material can stretch when pulled (please refer to ASTM D4964 for detailed definition, method for testing and measurement). The calculation method of strain for strain gauge is based on measuring the electrical signal that changes based on the stress (pressure) applied to the metallic cell on the strain gauge, so the concept is slightly different.

  1. Line 223: rather "are subjected" not "are subject". "to higher

tension due to stress intensity factor" - no factor will change the

strain, stress or deformation.

Line 223 has been revised accordingly

  1. Figure 2. The explanation of Lw and lc is debatable. If we assume a

certain initial state, then extending the connection by a certain value

of x should change both lengths by x (simplified). If we use more

advanced dependencies, they should be described in detail.

Lw is the change in the length of the waterproofing layer, while Lc is the length of movement of the concrete (either a crack or joint, commonly occurring due to freeze-thaw effect or vibration, depending on the structure and surrounding environment). While the change in the Lc may be predictable depending on the size of the crack/joint and the environmental factors involved, the change in the length of the waterproofing layer (Lw) would depend on the type of the waterproofing layer. If the concerned waterproofing layer is comprised of a cementitious material, in this case as the reviewer mentions a constant of x may be applied to both Lw and Lc, where as, for example, for certain compositely structured waterproofing sheets, there is barely any change to Lw. To consider all possible cases, the variables of Lw and Lc have been used (there are labels used for the purpose of explaining the concept limited to waterproofing materials and the degradation mechanism mentioned in the manuscript, and is not meant to be interpreted in a conventional mathematical common sense, and is a more simpler concept). Strain to deformation ratio can be a constant for concrete materials due to its well defined modulus of elasticity properties (of course, dependent on the national standards), but such is not the case for waterproofing materials, as there are far too many types. The advanced dependencies as the reviewer describes would be complicated to list in its entirety for this figure, but explanations have been supplemented in the text Lines: 168-176 in the revised manuscript

  1. I still don't know what the "strain concentration ratio" is. There is

no definition in the mathematical sense. The article presents the results of the experiment. However, if a definition of "strain concentration ratio" does not appear, they will be difficult to understand.

Strain concentration ratio is not meant to be defined in a mathematical sense, and is a term used to define the ratio of difference of strain 1) occurring at the center of the waterproofing layer/material above where the concrete crack/joint is moving to 2) the strain occurring at the local sections surrounding at center area, and this idea is limited to the scope of the study in the manuscript. This term was just coined to illustrate that entire concept above with a single phrase (if need be, can be changed to a simpler term). Please refer to Lines 181 to 184 in the revised manuscript for more clarified definition of this term.

  1. Figure 4. How far from the connection are the strain gauges? In

figure 4 (on the right) looks as if the strain gauges were not mounted

on the adhesive layer, but directly on the specimen.

Figure 4 has been revised to better explain the strain gauge attachment onto the specimen. Connection distance (?) from the strain gauges to the data acquisition system is not relevant as the connection is via wire. The data acquisition speed/rate would not be affected to a significant degree with just variable units of meters.

  1. What is the point of giving the equation (2)? And if I understand

correctly - giving resistance units incorrectly? If we are talking about

electresistant strain gauges, the resistance will be in ohms ... (?) Or

I don't know what the authors mean.

Equation 2 has been removed from the paper